# Variation and Predictors of Gross Motor Coordination Development in Azorean Children: A Quantile Regression Approach

**DOI:** 10.3390/ijerph19095417

**Published:** 2022-04-29

**Authors:** Sara Pereira, Flávio Bastos, Carla Santos, José Maia, Go Tani, Leah E. Robinson, Peter T. Katzmarzyk

**Affiliations:** 1Centre of Research, Education, Innovation and Intervention in Sport (CIFI2D), Faculty of Sport, University of Porto, 4200-450 Porto, Portugal; carlaaa_santos@hotmail.com (C.S.); jmaia@fade.up.pt (J.M.); 2Research Center in Sport, Physical Education, and Exercise and Health (CIDEFES), Faculty of Physical Education and Sports, Lusófona University, 1749-024 Lisboa, Portugal; 3Motor Behavior Laboratory, School of Physical Education and Sports, University of São Paulo, São Paulo 05508-030, Brazil; bastosfh@usp.br (F.B.); gotani@usp.br (G.T.); 4Child Movement, Activity, & Developmental Health Laboratory, School of Kinesiology, University of Michigan, Ann Arbor, MI 48109, USA; lerobin@umich.edu; 5Pennington Biomedical Research Center, Baton Rouge, LA 70808, USA; peter.katzmarzyk@pbrc.edu

**Keywords:** motor skills, youth, mixed models, median regression

## Abstract

We investigated the development of gross motor coordination (GMC) as well as its predictors in school-aged Azorean children. The sample included 181 children (90 girls), followed consecutively for 4 years from 6 to 9 years of age. GMC was assessed with the Körperkoordinationstest für Kinder, and predictors included body mass index, standing long jump, 50-yard dash, and shuttle run. The changes in GMC and the effects of predictors were analyzed with mean-modeling as well as quantile regression. In the latter, we considered the following three quantiles (Q): Q20, Q50, and Q80 as markers of low, median, and high GMC levels, respectively. All analyses were conducted using R software and alpha was set at 5%. The GMC changes were curvilinear in both models, but the quantile approach showed a more encompassing picture of the changes across the three quantiles in both boys and girls with different rates of change. Further, the predictors had different effect sizes across the quantiles in both sexes, but in the mean-model their effects were constant. In conclusion, quantile regression provides more detailed information and permits a more thorough understanding of changes in GMC over time and the influence of putative predictors.

## 1. Introduction

In 1974, Kiphard and Schilling [1], two German scholars, defined gross motor coordination (GMC) as the harmonious and economic interactions of the neuromuscular and sensory systems to produce precise and balanced motoric actions, as well as adequate reactions to a varied set of situations, and they also developed a test battery to assess GMC—the *Körperkoordinationstest für Kinder* (KTK). Two years later, the battery was presented to an English-speaking audience [2] and gave rise to the first study outside Germany that investigated the effects of decision making on motor skills and self-concept in US children [3]. In 1980, the KTK entered the mainstream in a motor development book [4] and, in the same year, the first longitudinal GMC study using the KTK was published [5]. A review of the GMC assessment has recently been conducted [6], as well as its relatedness to other available instruments to assess movement skills [7]. Currently, promoting the development of GMC has been considered a key strategy during childhood when the objective is to encourage health-related quality of life, including long-term obesity prevention [8] and the maintenance of cardiorespiratory fitness [9]. It has been found that GMC is positively related to higher levels of physical fitness [10], increased moderate-to-vigorous physical activity [11], and aspects of children’s psychological development, i.e., cognition and intelligence [12].

Nevertheless, since Willimczik’s longitudinal paper in 1980 [5], the relevance of relating serial GMC data to different aspects of children’s multifaceted motoric development and its links to their active and healthy lifestyles laid dormant for about 30 years. Then, from 2010 onwards [13], we witnessed a resurgence of longitudinal research using the KTK test battery to describe children’s GMC development and its putative predictors. This trend has two main streams of investigation and available data comes primarily from Portugal, Brazil, Belgium, and Denmark. The first stream used GMC as an independent variable to examine its association with sports participation, physical activity, fitness, BMI, and body fat [8,14,15,16,17]. The second stream focused on describing trends in serial GMC data (as a dependent variable) and examined the additive effects of different predictors on its mean trajectory. For example, Deus et al. [13] modeled mean changes in each of the four KTK tests as a function of chronological age, BMI, and physical activity. D’Hondt et al. [18] investigated how different mean changes in GMC were associated with weight status (i.e., overweight versus normal weight) in Belgium children, and dos Santos et al. [19] used an allometric approach to identify key modifiable determinants of GMC in Azorean children. Furthermore, Reyes et al. [20] studied the dynamics of sex differences in Portuguese mainland children’s GMC and its correlates, such as birth weight, BMI, laterality, fitness, physical activity, and socioeconomic status. Nonetheless, if gross motor coordination comprises the ability to use and control our neuromuscular system to perform certain movements with efficiency, then velocity, agility, and explosive leg strength are important components given that they are linked to gross motor coordination performance levels in children and youth. Yet, to the best of our knowledge, only a few studies examined the influence of these components as a set, on GMC trajectories. One example is Reyes et al. [20], which showed that stronger and more agile children were more likely to display higher GMC developmental trends.

Notwithstanding the relevance of the longitudinal studies listed above, it is important to stress that they concentrated their analyses on KTK mean differences, or modeling KTK mean trends conditional on fixed and dynamic predictors which are assumed to have a constant effect across age. In a sense, these studies apparently followed Ulrich’s [21] definition of motor development, focusing on typical trajectories as well as the factors that influence motor behavior. Yet, these longitudinal reports do not provide a more encompassing picture regarding other possible GMC trajectories beyond the mean. Following Koenker and Hallock [22], we also concur with Mosteller and Tukey [23] that we should further expand our thinking about going beyond the traditional focus on the mean. For example, the percentile charts of KTK tests for Portuguese boys and girls are available and provide a broader description of GMC performance levels far beyond sample means. In walking backwards along a balance beam, for example, for both Azorean boys and girls, Maia and Lopes [24] reported the following results: for 6-year-old boys, percentile 3 (P_3_) = 5.5, P_50_ = 22.5, and P_97_ = 50.1 points, whereas for 9-year-olds, P_3_ = 14.9, P_50_ = 42.3, and P_97_ = 68.1 points; for 6-year-old girls, P_3_ = 6.3, P_50_ = 25.2, and P_97_ = 56.7 points, whereas for 9-year-olds, P_3_ = 15.0, P_50_ = 41.5, and P_97_ = 67.1 points. Similar trends are shown in the other three tests, and analogous data were reported for boys and girls from mainland Portugal [25] and the Madeira islands [26]. This, of course, challenges the incomplete information about the distribution of children’s GMC when the concentration is exclusively on the mean, i.e., the typical trajectory, as well as the constancy of the fixed effects of its predictors.

Koenker and Basset [27] introduced quantile regression to come to terms with the limitations of conditional mean-modeling, and although there are books on the subject [28,29], and Petscher and Logan [30] provided a primer for developmental researchers, to our knowledge it has not been used to study GMC developmental trajectories. We contend that it would be of interest to verify whether the results from quantile regression would be similar, or different, from those obtained from the conditional-mean model. If so, these would be very important for physical education teachers as well as youth sports’ coaches.

More specifically, given children’s systematic differences in their motor development and, therefore, in their dissimilar gross motor coordination trajectories, quantile regression results can help teachers and coaches develop more assertive programs and interventions based on children’s similarities and differences, especially in their varied developmental levels. Truly, children are more different than alike, and this has to be strongly grasped by PE teachers and coaches. Furthermore, the available longitudinal results from conditional-mean models previously mentioned did not provide estimates for rates of change across children’s age when GMC shows a curvilinear trend. On the other hand, when using quantile regression, children’s GMC trajectories may have different rates of change at each point of the distribution. This is of importance to have a broader understanding of children’s distinct GMC trajectories, when planning physical education interventions in primary school children aimed at improving their motor coordination. Therefore, this study aims to: (1) describe longitudinal changes in children’s GMC; (2) determine the effect sizes of BMI, explosive leg strength, running speed, and agility in predicting GMC; (3) estimate rates of change in GMC across age. Therefore, we hypothesize the following: (1) GMC will show a curvilinear trend in both boys and girls; (2) BMI, agility, and explosive leg strength will significantly correlate with GMC development; (3) GMC quantiles will unfold differently in boys compared to girls; (4) their correlates will show distinct effect sizes; (5) GMC rates of change will be different in the mean-modeling compared to the quantile modeling approach.

The aims will be accomplished using the following two approaches: the first is mean-modeling and the second is quantile regression. Both approaches will provide data on how GMC changes over time. Yet, whereas the first approach only models mean GMC as a function of age, BMI, explosive leg strength, running speed, and agility, the second models sets of individuals who change their GMC in different ranks, i.e., quantiles of the GMC distribution, and how BMI, explosive leg strength, running speed, and agility might differently affect their GMC development. In the present paper, we will only consider three quantiles: Q_20_ expressing low GMC levels, Q_50_ showing the typical (median) trajectory, and Q_80_ displaying high GMC developmental trajectory.

## 2. Materials and Methods

### 2.1. Sample

Our sample comes from the Azores archipelago, a Portuguese autonomous region located in the Atlantic Ocean (between 36.5–40° North latitudes, and 24.5–31.5° West longitudes), whose main employment sectors are services, agriculture, fishery, industry, and tourism [31]. The archipelago has nine islands. Of these, the most important in terms of population size and number of schools are S. Miguel, Terceira, Faial, and Pico, comprising ~80% of the total school-aged population [32].

In 2002, we initiated a research project using a mixed-longitudinal design aiming to study the dynamics of physical growth, body composition, biological maturation, GMC, physical activity, physical fitness, motivation for sports participation, and socioeconomic status within the context of their school settings and islands [24]. Data were collected from four age cohorts with a one-year overlap such that in cohort 1 subjects were followed consecutively from 6 to 10 years of age, in cohort 2 from 10 to 13, in cohort 3 from 13 to 16, and in cohort 4 from 16 to 19 years of age, and all assessments were done between September and October. The projected sample was randomly collected in a stratified manner from each island, aiming to have 250 individuals per cohort, totaling 1000 individuals per year.

For the present study, we will only use data from cohort 1. Although children from both sexes were followed from 6 to 10 years of age, GMC was only assessed from 6 to 9 years of age, covering the primary school years. In total, we have 181 individuals (90 girls) with complete data on all assessments. No statistically significant mean differences (*p* < 0.05) were observed between the children included in the present article from those excluded with missing data. Informed consent was obtained from all parents and the study was approved by the Secretary of Education of the Azorean Regional Government, as well as by the ethics committee of the Faculty of Sport, University of Porto, Portugal.

### 2.2. Anthropometry

Following standard protocols established by the International Society for the Advancement of Kinanthropometry [33], stature was measured with children in an upright position and with their head to the Frankfurt plane with a portable Siber Hegner stadiometer (Siber Hegner, Zurich, Switzerland) to the nearest 0.1 cm. Body mass was measured with children with light clothing and without shoes on a Seca scale (Seca Optima 760, Germany) to the nearest 0.1 kg. Body mass index (BMI) was calculated using the standard formula: BMI = body mass (kg)/stature (m^2^).

### 2.3. Gross Motor Coordination

Gross motor coordination was assessed with the *Körperkoordinationstest für Kinder* test battery (1, 2) for children aged 5.00–14.99 years of age, comprising the following four tests.
Walking backwards (WB) along a balance beam assesses dynamic balance. Children walk backwards on three balance beams with different widths (6 cm, 4.5 cm and 3 cm), all with 3 m length and 5 cm height. The total number of successfully completed steps (maximum of 8) in three trials for each balance beam was registered.Hopping on one foot (HOF) is a measure of lower limb coordination and dynamic energy/strength. Participants hopped over one pillow (60 cm × 20 cm × 5 cm each) without touching, and then hopped on the same foot at least 2 times to have a successful attempt. After each successful attempt, one pillow was added (maximum of 12 pillows). Children had three attempts on both feet at each level. If participants did not have success after the three attempts, the test finished. A successful hop on the first trial receives 3 points, on the second trial, receives 2 points, and on the last trial, receives 1 point, totalizing a maximum of 78 points that could be reached (39 per leg).Jumping sideways (JS) marks speed in alternated jumping. Participants jumped laterally with both feet over a wooden slat (60 cm × 4 cm × 2 cm), as fast as possible for 15 s. Two trials were completed, and the total number of successful jumps was recorded.Moving sideways on boxes (MSB) assesses laterality, as well as a space-time structure. Participants stand with both feet on a platform (25 cm × 25 cm × 5 cm) and place both hands on an adjacent platform (with the same dimensions). They then place the second platform alongside the first and step on it. This movement is repeated as fast as possible during a 20 s trial (2 trials). The total number of relocations (2 points) per trial was recorded and summed. One practice attempt was allowed in each test so that the participant could become familiar with the aim of each test.

A total KTK score (GMC_T_) was obtained by summing the scores from each test as suggested [34]. Recent factor analysis confirmed that the KTK is a viable way to capture the overall gross motor coordination across the four tests [35]. Further, the KTK battery has been translated into Portuguese and has been widely used with Portuguese children [20,36,37,38].

### 2.4. Performance-Related Physical Fitness

Three tests from the American Alliance for Health, Physical Education, Recreation, and Dance (AAHPERD) youth test battery [39] were used as follows. The standing long jump test measures explosive leg strength. Children stand behind a marked line, with their feet apart. After swinging their arms and bending their knees, they are expected to jump as far as possible, and the resulting performance is expressed in meters. The average of three trials was considered for analysis. The 50-yard dash (45.7 m) measures running speed. Children are instructed to run this distance as fast as possible, with the resulting performance expressed in seconds. The average of two trials was considered for analysis. For the present study, it will be expressed in meters per second (m·s^−1^) since it is the most adequate way to express speed. The shuttle run test marks agility, emphasizing speed and change of direction. Children are expected to run as fast as possible between two parallel lines 9 m apart. Two small wood blocks were placed behind one line. Then, children run as fast as possible, pick up one wood block, run back to the starting line and place the block behind the line, and repeat the task of retrieving the second block. The average of two trials was considered for analysis. Although the resulting performance is expressed in seconds, for the present study it will be expressed in meters per second (m·s^−1^), because of the speed component in this test.

### 2.5. Data Quality Control

Data quality control was performed in a series of steps, as follows: (1) team members were trained by the principal investigator to reduce assessment errors; (2) a series of data registries was built in FileMakerPro v. 5.0 (Claris, International Inc., Cupertino, CA, USA) to facilitate data entry, as well as control for punching errors and out of range values for each variable; (3) a sample of 25 children (13 boys) from each of the four islands was randomly selected for reliability estimation; (4) test-retest ANOVA-based intraclass correlation coefficients (R) were calculated as follows: R = 0.98 for stature and R = 0.99 for body mass in both boys and girls; GMC tests ranged from R = 0.75 in MSB to R = 0.91 in JS in boys, and from R = 0.79 in WB to R = 0.91 in MSB in girls. In boys, reliability estimates were R = 0.98 for standing long jump, shuttle-run, and 50-yard dash; for girls, R-values ranged from 0.94 (50-yard dash) to 0.98 in the shuttle-run.

### 2.6. Data Analysis

The descriptive statistics are presented as means and standard deviations. To comply with the study aims, the analysis strategy was divided into two steps. In step 1, GMC_T_ serial data, within each sex, were analyzed with a linear-mixed model (mean-modeling) [40]. Here, a second-degree polynomial of age was fit to the data to best capture the curvilinear trend in GMC_T_, and then predictors were added. In sum, the goal was to predict mean changes in GMC_T_ as a function of age, BMI, and motor performance indicators (standing long jump, shuttle run, and 50-yard dash). The module *lme4* of R software was used in these computations, and all parameters were simultaneously estimated using maximum likelihood [41,42]. In step 2, GMC_T_ serial data was submitted to a linear-mixed quantile regression approach, a suitable method to deal with random intercepts and slopes of individuals located in different quantiles at each age, and all parameters were simultaneously estimated using the asymmetric Laplace likelihood [43]. Given the sample size, and as previously mentioned, we only considered the following three quantiles (Q) that mark distinct GMC developmental trajectories: low (Q_20_), median or typical (Q_50_), and high (Q_80_). All computations were performed with the *lqmm* module developed and implemented in R software by Geraci [44], and Geraci and Bottai [43]. As usual, in both steps, the predictors were centered at the grand mean [45]. Since we have a second-degree polynomial, the GMC_T_ rates of change (Δcg) were calculated at each age using the following formula: Δcg = β_1_ + 2β_2_ (age). Further, as advocated [40], age was anchored at 6 years. In all analyses, the significance level was set at 5%.

## 3. Results

Descriptive statistics are presented in Table 1, and as expected, there are systematic mean increases in GMC_T_ as well as in BMI and physical fitness tests in both sexes across age. Apart from an apparent distinction in sex-differences in GMC_T_ means across age, the other variables did not show a clear tendency favoring either sex.

Parameter estimates related to modeling the mean as well as the three different quantiles (Q_20_, Q_50_, Q_80_) are presented in Table 2 for both boys and girls. In the mean model approach, boys showed curvilinear trends in GMC_T_ (age and age^2^ are statistically significant). At 6 years of age, boys’ GMC_T_ is 112.50 ± 2.89 points. Further, on average, there is a negative association between GMC_T_ change and BMI (β = −2.94 ± 0.70, *p* < 0.05), and a positive association with standing long jump (β = 15.59 ± 5.61, *p* < 0.05), shuttle run (β = 14.33 ± 4.59, *p* < 0.05), and the 50-yard dash (β = 9.17 ± 2.67, *p* < 0.05). On the other hand, quantile regression shows different results. Six-year-old boys have distinct GMC_T_ values, i.e., at Q_20_ = 106.72 ± 2.53 points, at Q_50_ = 119.29 ± 2.56 points, and at Q_80_ = 133.73 ± 2.67 points. Furthermore, whereas the GMC_T_ trend is linear at Q_20_, it has a curvilinear trend in Q_50_ and Q_80_. BMI, shuttle-run, and the 50-yard dash distinctively affect GMC_T_ across the three quantiles. For example, the mean model showed that, on average, the association of the shuttle run with GMC_T_ change is 14.33 ± 4.59 points, whereas in Q_20_ it is 18.17 ± 7.55 points, and in Q_50_ and Q_80_ no significant relation with GMC_T_ change was found. On the contrary, the 50-yard dash was associated differently with each quantile (Q_20_ = 16.51 ± 5.12 points, Q_50_ = 16.38 ± 4.94 points, Q_80_ = 21.12 ± 4.90 points).

In girls, a relatively analogous picture emerges. For a random 6-year-old girl, the estimated GMC_T_ is 108.42 ± 3.12 points in the mixed-model, whereas for those at Q_20_ it is 104.61 ± 3.28, at Q_50_ = 115.72 ± 3.40, and at Q_80_ = 131.53 ± 3.38. Moreover, the mean model shows a curvilinear trend in GMC_T_ development, but in Q_20_ and Q_50_ it is linear; yet, in Q_80_ the trend is curvilinear. BMI and shuttle-run do not associate with GMC_T_ changes in the mean model but do so in the quantile model with different effect sizes, except for BMI in Q_80_. Similarly, for the 50-yard dash, the mean model shows an effect of 15.45 ± 3.49 points on GMC_T_ change, Q_20_ = 18.86 ± 3.55 points, Q_50_ = 13.94 ± 3.78 points, and Q_80_ = 15.47 ± 3.81 points.

Figure 1 shows the three quantiles’ (Q_20_, Q_50_, and Q_80_) trajectories across age for both boys and girls embedded in intra-individual trajectories (spaghetti plot).

Table 3 shows GMC_T_ rates of change for both boys and girls. Findings reveal distinct results not only when comparing the mean model with the quantile regression model, but also across quantiles. In boys, for example, the mean model showed a consistent decrease in the rate of change, from 29.93 points·y^−1^ at 6 years of age to 11.29 points·y^−1^ at 9 years of age. In Q_20_, the rate of change is constant, i.e., 29.93 points·y^−1^, whereas in Q_50_ rates are different from the mean model (from 24.24 points·y^−1^ at 6 years to 6.84 points·y^−1^ at 9 years), and in Q_80_ differences are even more marked across the entire age range, from 26.26 points·y^−1^ at age 6 to 1.94 points·y^−1^ at age 9. In girls, a similar picture emerges. Although the mean model shows a consistent decline in rates of change, from 26.55 points·y^−1^ at 6 years of age to 16.17 points·y^−1^ at 9 years, the quantile model reveals a different trend, i.e., in Q_20_ and in Q_50_, the rate of change is constant (13.87 points·y^−1^ and 19.85 points·y^−1^, respectively), but in Q_80_, the decline is more marked than shown by the mean model, ranging from 23.73 points·y^−1^ at 6 years of age to 11.19 points·y^−1^ at 9 years of age.

## 4. Discussion

In summary, we showed that quantile regression offers a more encompassing representation of the GMC developmental fabric than the mean-modelling approach. Given the intricacy of our study and its results, we will address the discussion in the following two steps: first, on methodological grounds (gross motor coordination assessment and use of quantile regression); second, on substantive terms to interpret what we found.

### 4.1. Methodological

Assessing gross motor coordination in children and adolescents is probably far more challenging and complex than meets the eye. Further, each assessment tool conveys the expectation of its use in pedagogical (physical education and sports participation) and rehabilitation (clinical) settings. Two such putative reliable and valid tools are the KTK test battery [1] and the Movement Assessment Battery for Children (M-ABC) [46]. Each provides not only normative and criterion-related assessments, but also an overall index so that children and adolescents can be classified as with normal coordination or with coordination impairments. Although the M-ABC test battery also captures manual dexterity, ball skills, and static and dynamic balance, the KTK relies on a single-factor solution termed gross motor coordination. This structure may be a limitation in itself because it does not consider manipulative tasks, although moving sideways in boxes require some form of manipulation. Yet, the KTK is very easy to administer, is not time consuming, and the results’ interpretation is easy to grasp. Further, it has systematically been used in educational settings [47,48,49] as well as in youth sports [7,50].

Another important methodological issue to tackle is: how much better and more encompassing are quantile regression results than those provided by the mean-modeling approach [51]? There is no doubt that available reports using the mean-modeling methodology with either cross-sectional or longitudinal data provided important material on how GMC unfolds during childhood and adolescence. However, one has to bear in mind the following two important issues [43]: first, that we are only modeling mean GMC changes, and second, that derived explanatory variables’ coefficients for BMI, explosive leg strength, agility, and running speed would each have a constant effect irrespective of the position of children’s developmental trends in their entire distribution. This apparently is not accurate because we showed that GMC trends can, and most probably should, be examined on specific sub-groups at quantiles of interest (Q_20_, Q_50_, Q_80_), and that predictors of change have different effect sizes on these sub-groups’ distinct trajectories, providing a more detailed interpretation of the unfolding complexities of GMC and some of its correlates.

There is apparently “missing reporting” in existing studies using the mean-model to investigate GMC curvilinear trends since they almost never presented rates of change across age. If available, these could be of primary interest for a differential interpretation of change at different ages. Yet, even if these results were presented, the model assumes that the rate of change is the same for all children across the entire distribution at that age. Yet, with the quantile approach, a different picture emerged, not only at each age, but also at different quantiles of the distribution, i.e., in different sub-groups of children. This is the first time that such information is provided for GMC.

### 4.2. Substantive

Cross-sectional [38,52] and longitudinal [18,53] reports consistently showed that with increasing age, GMC also increases, although differences between adjacent ages (e.g., 5 to 6 yrs, 6 to 7 yrs, 7 to 8 yrs, etc.) are not of the same magnitude and these are apparently sex-specific. This most probably reflects the putative effects of differences in children’s fundamental motor skill development, as well as their physical fitness and physical activity levels [52]. The mean-model approach—arguably focusing on the typical trajectories of GMC development—captured a curvilinear trend for all boys and girls, confirming previous findings [13,20], and the median value (P_50_), as shown in percentile charts’ representations for each of the KTK tests [25,26]. Yet, the quantile regression results provided an unexpected developmental trend—boys in the lower developmental canal (Q_20_) showed a linear GMC trend, whereas boys in the median (Q_50_) and higher (Q_80_) canals exhibited a curvilinear trend. In turn, only girls in the higher canal showed a curvilinear GMC trend. One could argue that participants in the higher GMC quantiles could be reaching a ceiling effect. Thus, by approaching the ceiling—i.e., the maximum summed score possible—a reduction in GMC velocity would be observed. In this case, the reduction could be caused by either a test limitation or by the GMC dynamics in distinct GMC levels. Critically, descriptive analysis does not suggest participants reaching an asymptote, which does not corroborate the ceiling effect hypothesis. Nevertheless, future studies could further investigate if there are children approaching a GMC asymptote at around 10 years old.

Our results indicate that the rates of change across age using the quantile approach reveals a new perspective over GMC development compared to previous reports. The mean-modeling approach, in line with the search for a typical trajectory, reveals that the GMC development of a “typical child” unfolds at progressively smaller steps across time. In turn, the quantile approach indicates that while GMC develops at a constant rate for boys and girls with lower levels of GMC, children within higher trends of GMC show a decreasing rate of change across time. This result suggests that the rate of change may be negatively associated with how much one can still improve in GMC if they have access to appropriate motor experiences within favorable contexts. However, unexpectedly, girls in higher GMC trends (Q_80_) until 8 years old show higher rates of change compared to girls in the lower quantile (Q_20_), which does not occur for boys. One possible explanation may be linked to the differences in mean values between boys and girls. In fact, descriptive analysis suggests that girls may have lower mean values, compared to boys, which is consistent with cross-sectional and short-term longitudinal reports [47,54], longitudinal data from Vouzela children [20], and from a recent systematic review [55]. Thus, one could argue that the hypothesis of a higher rate of change for those with more room for improvement could likely explain this pattern of results for girls. Nevertheless, at this point, this suggestion is merely speculative.

We expected that approaching GMC development from a quantile perspective would reveal that predictors distinctly contribute to GMC, depending on the level of GMC itself. In fact, our results show that the shuttle-run and standing long jump tests can have distinct contributions to GMC, conditional on children’s GMC trajectory. Specifically, agility, measured by the shuttle-run test, contributes strongly to the GMC unfolding of boys with lower GMC. From a mean-model perspective, a strong contribution of this predictor for GMC development would conceal the absence of a relationship for boys within median and higher GMC trajectories. Furthermore, agility has also been shown to strongly contribute to the GMC development of girls across GMC levels. Nevertheless, the mean-model approach failed to reveal this association. With respect to explosive leg strength, assessed by the standing long jump test, our results indicate that its contribution to GMC across time occurs independently of the analytical approach for both boys and girls. Nevertheless, the difference in the predicted effect sizes for GMC is not negligible between the mean-model approach and the quantile regression model, with a larger contribution predicted by the last (for instance, mean-model: 15.59 and Q_80_: 49.04). Running speed shows a similar association to GMC unfolding, compared to what was observed for the explosive leg strength. Nevertheless, in this case, larger effect sizes were observed only for boys (for instance, 9.17 in the mean-model, compared to 21.12 for the Q_80_ in the quantile regression), with girls showing similar effect sizes between both analytical approaches.

One of the key points highlighted in the Barnett, Lai, Veldman, Hardy, Cliff, Morgan, Zask, Lubans, Shultz, Ridgers, Rush, Brown, and Okely [55] systematic review refers to how correlates of motor competence can be distinct, depending on how they were assessed. In this sense, the present study adds to the methodological challenges of studying GMC correlates, including that their contribution depends also on the chosen analytical approach—i.e., mean-model or quantile regression. Additionally, even if a decision is made to approach GMC unfolding by a quantile perspective, correlates can depend also on the GMC level. For instance, using shuttle-run and standing long jump tests as physical fitness predictors, Coppens et al. [56] found that, although both explained the intercept variance, neither was significantly associated to GMC unfolding over time. Our results suggest that the intricate relationship between GMC and these predictors can be explained by the distinct association they have shown conditional on the GMC level.

The model proposed by Stodden et al. [57] predicts a negative association between BMI and GMC development. A recent integrative review on pediatric obesity [58] corroborates the model prediction, indicating that the available evidence points to an association between obesity and motor deficits. Our results also show that BMI is associated with GMC development, independent of GMC levels in boys. Critically, the same is not true for girls. Specifically, the mean-model approach does not indicate a significant relationship between BMI and GMC change in girls. In turn, BMI has shown to associate with GMC unfolding for girls presenting lower and median GMC levels (Q_20_ and Q_50_). One could argue that our participants’ BMI does not cover the range from thinness to obesity (i.e., does not include both normal weight children and those with obesity), which would result in a poor prediction regarding the contribution of BMI to GMC change across time (e.g., at the age of 6 years old, girls: 17.33 ± 2.72, boys: 17.17 ± 2.53). Using the WHO [59] growth data as reference, our sample indeed has zero girls and boys below the cut-off for thinness. Nevertheless, 16 girls (17%) and 18 boys (20%) are above the cut-off for obesity, which provides data points within this range for prediction.

Notwithstanding the relevance of our results, this study is not without limitations. First, the sample is from an Autonomous Region of Portugal—the Azores archipelago—which limits the generalization to all Portuguese children. Second, we do not have information on biological maturation, and it is possible that this could have an effect on children’s GMC unfolding. Although we have recently reported, with cross-sectional data, that skeletal age only has a negligible effect on GMC levels in children [60], it is open for future research to probe into its putative differential effects on gross motor coordination development. Third, we do not have clinical or developmental justifications for using the three quantiles and their labeling—low, median or typical, and high GMC levels. Yet, we believe the reader will not disagree with us that such quantiles may define, confidently, the distinct GMC developmental levels.

## 5. Conclusions

In conclusion, although the mean model showed a curvilinear trend in both boys’ and girls’ GMC development, quantile regression revealed a more encompassing developmental picture depending on where children are in their quantiles (Q_20_, Q_50_, or Q_80_). Further, though shuttle run, standing long jump, and the 50-yard dash have different, but constant effect sizes in GMC unfolding in boys and girls, a different developmental trend emerges in the three quantiles in both boys and girls. Finally, whereas the GMC rates of change decline with age in both boys and girls, in the quantile approach this decline is different in Q_20_, Q_50_, and Q_80_.

In sum, we showed that to best understand how children’s GMC unfolds, a more encompassing statistical model that goes beyond the mean-modeling approach is probably needed. Quantile regression provides more detailed information on how children change with age and permits a thorough understanding for developing effective interventions by physical education teachers and sports coaches because children are more different than alike, not only in their coordination readiness, but also in their developmental potential.

## Figures and Tables

**Figure 1 ijerph-19-05417-f001:**
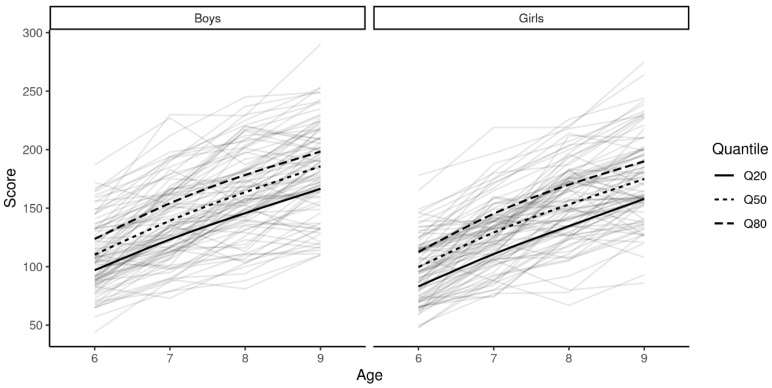
Modeled trajectories of the three percentiles, Q_20_, Q_50_, and Q_80_, in boys and girls.

**Table 1 ijerph-19-05417-t001:** Means ± standard deviations of all variables in both boys and girls across age.

	Age (In Years)			
Variables	6 ^§^	7	8	9
	Boys			
GMC_T_ (points)	108.35 ± 29.72	138.86 ± 34.43	162.45 ± 37.80	182.73 ± 29.82
Body mass index (kg·m^−2^)	17.33 ± 2.72	17.30 ± 3.06	17.83 ± 3.42	18.60 ± 3.80
Standing long jump (m)	0.93 ± 0.16	1.04 ± 0.20	1.08 ± 0.20	1.22 ± 0.22
Shuttle run (m·s^−1^)	2.53 ± 0.22	2.56 ± 0.25	2.69 ± 0.27	2.83 ± 0.30
50-yards dash (m·s^−1^)	3.96 ± 0.44	4.15 ± 0.38	4.47 ± 0.44	4.78 ± 0.58
	Girls			
GMC_T_ (points)	98.21 ± 27.58	127.84 ± 29.28	156.16 ± 34.09	174.24 ± 37.71
Body mass index (kg·m^−2^)	17.17 ± 2.53	17.17 ± 2.76	17.88 ± 2.96	18.66 ± 3.34
Standing long jump (m)	0.86 ± 0.14	1.01 ± 0.20	1.04 ± 0.17	1.10 ± 0.20
Shuttle run (m·s^−1^)	2.43 ± 0.25	2.47 ± 0.23	2.63 ± 0.25	2.68 ± 0.31
50-yards dash (m·s^−1^)	3.66 ± 0.39	3.96 ± 0.36	4.19 ± 0.42	4.39 ± 0.47

^§^ In terms of chronological age, 6 years refers to all children aged 6.00 to 6.99 in decimal years, and the same occurs for 7, 8, and 9 years.

**Table 2 ijerph-19-05417-t002:** Parameter estimates (± standard errors) for all subjects using mean-modeling and estimates for the three quantiles–Q_20_, Q_50_, Q_80_ in both boys and girls.

	Modeling the Mean	Modeling Quantiles		
	Parameter Estimates ± s.e.	Parameter Estimates ± s.e. Q_20_	Parameter Estimates ± s.e. Q_50_	Parameter Estimates ± s.e. Q_80_
		Boys		
Intercept	112.50 ± 2.89	106.72 ± 2.53	119.29 ± 2.56	133.73 ± 2.67
Age	28.93 ± 2.57	20.93 ± 3.73	24.24 ± 3.03	26.36 ± 3.50
Age^2^	−2.94 ± 0.82	−2.74 ± 1.18 ^ns^	−2.90 ± 1.13	−4.07 ± 1.25
Body mass index (kg·m^−2^)	−2.03 ± 0.70	−1.35 ± 0.62	−1.80 ± 0.75	−1.83 ± 0.82
Standing long jump (m)	15.59 ± 5.61	49.26 ± 10.59	47.64 ± 10.33	49.04 ± 10.62
Shuttle run (m·s^−1^)	14.33 ± 4.59	18.17 ± 7.55	13.52 ± 8.00 ^ns^	10.78 ± 7.87 ^ns^
50-yard dash (m·s^−1^)	9.17 ± 2.67	16.51 ± 5.12	16.38 ± 4.94	21.12 ± 4.90
		Girls		
Intercept	108.42 ± 3.12	104.61 ± 3.28	115.72 ± 3.40	131.53 ± 3.38
Age	26.55 ± 2.88	13.87 ± 3.56	19.85 ± 3.55	23.73 ± 3.17
Age^2^	−1.73 ± 0.85	−0.52 ± 1.06 ^ns^	−0.49 ± 0.94 ^ns^	−2.09 ± 0.78
Body mass index (kg·m^−2^)	−2.52 ± 0.71 ^ns^	−1.65 ± 0.85	−2.81 ± 0.91	−1.62 ± 0.82 ^ns^
Standing long jump (m)	13.55 ± 6.33	50.33 ± 10.38	42.78 ± 10.46	48.32 ± 10.12
Shuttle run (m·s^−1^)	8.71 ± 5.03 ^ns^	19.49 ± 5.81	14.91 ± 6.21	17.32 ± 5.67
50-yard dash (m·s^−1^)	15.45 ± 3.49	18.86 ± 3.55	13.94 ± 3.78	15.47 ± 3.81

^ns^ = non-significant.

**Table 3 ijerph-19-05417-t003:** Gross motor coordination estimated rates of change (Δ_cg_) expressed as points·y^−1^ in both boys and girls with the mean-modeling versus the quantile approach.

Years	Mean-Modeling		Q_20_		Q_50_		Q_80_	
	Boys	Girls	Boys	Girls	Boys	Girls	Boys	Girls
6	29.93	26.55	29.93	13.87	24.24	19.85	26.26	23.73
7	23.05	23.09	29.93	13.87	18.44	19.85	18.28	19.55
8	17.17	19.63	29.93	13.87	12.64	19.85	10.08	15.37
9	11.29	16.17	29.93	13.87	6.84	19.85	1.94	11.19

## Data Availability

The data used in this study can be requested from one of the authors (J.M.) who is the curator of the broad data set of this study.

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
