# Peer review of "Variation and Predictors of Gross Motor Coordination Development in Azorean Children: A Quantile Regression Approach"

_ijerph, 2022, doi:10.3390/ijerph19095417_

Round 1
Reviewer 1 Report
Dear Author,
Thank you for your work. I have a few minor comments:
Please define how the sample size was defined.
Page 1 Line 34:
Usually, gross motor function is measured with: Gross Motor Function Classification System, Gross Motor Function Classification System-Expanded & Revised, Manual Ability Classification System, and Communication Function Classification System. What is the difference with KörperkoordinationsTest für ? Please comment on it.
Page 3 Line 100:
Please state the Hypotheses of the study.
Page 3 Line145:
State if the KörperkoordinationsTest für Kinder was translated in Portuguese.
Page 3 Line146:
Consider to present the tests with bullet points. Now, it Is difficult to follow the paragraph
Page 4 Line 209.
While the Statistical analysis is well present, there is no mention about the Power (Beta) and how the sample size was computed. Linear-mixed model are sample size sensitive. Please add if the sample size estimation was run.
Page 10 Line 385
Comment on the generalizability of the study findings. The population seems unique, i.e., Azores archipelago.
Thank you
Reviewer 2 Report
Please see the attached

Reviewer 3 Report
Title, Abstract and Discussion- Azores has a specific cultural and social context, that may influence motor development and gender differences in motor development; so, it is adviseble that title, abstract and discussion reflects this Portuguese regional specificity
lines 137-43- Anthropometry- include adequate references for all measures.
lines 346-353- For this age group, gender differences are becoming less and less evident in more recent, so we strongly advise authors to consider more recent studies and review own reflexions
lines 363-372- One of the critics we have to make is the absence of measures relative to maturation. At the end of this age group it is predictable that some boys and girls have reached puberty. So, results of tests that depend more on conditional capacities are more influenced by maturation; and your statistical results and fit models seem to support this hypothesis. So, we strongly recommend inclusion of study limitations, for this subjetc and others.
